# Antiviral Activity of Jamaican Medicinal Plants and Isolated Bioactive Compounds

**DOI:** 10.3390/molecules26030607

**Published:** 2021-01-25

**Authors:** Henry Lowe, Blair Steele, Joseph Bryant, Emadelden Fouad, Ngeh Toyang, Wilfred Ngwa

**Affiliations:** 1Biotech R & D Institute, University of the West Indies, Mona, 99999 Kingston, Jamaica; lowebiotech@gmail.com (H.L.); jbryant@ihv.umaryland.edu (J.B.); 2Vilotos Pharmaceuticals Inc., Baltimore, MD 21202, USA; ngeh.toyang@flavocure.com; 3Flavocure Biotech Inc., Baltimore, MD 21202, USA; 4Institute of Human Virology (IHV), University of Maryland School of Medicine, Baltimore, MD 21201, USA; 5Physics Department, Florida Polytechnic Institute, Lakeland, FL 33805, USA; efouad@floridapoly.edu (E.F.); wngwa@bwh.harvard.edu (W.N.); 6Brigham and Women’s Hospital, Dana-Faber Cancer Institute, Harvard Medical School, Boston, MA 02215, USA

**Keywords:** phytomedicine, viral infections, antivirals, phytoantiviral

## Abstract

Plants have had historical significance in medicine since the beginning of civilization. The oldest medical pharmacopeias of the African, Arabian, and Asian countries solely utilize plants and herbs to treat pain, oral diseases, skin diseases, microbial infections, multiple types of cancers, reproductive disorders among a myriad of other ailments. The World Health Organization (WHO) estimates that over 65% of the world population solely utilize botanical preparations as medicine. Due to the abundance of plants, plant-derived medicines are more readily accessible, affordable, convenient, and have safer side-effect profiles than synthetic drugs. Plant-based decoctions have been a significant part of Jamaican traditional folklore medicine. Jamaica is of particular interest because it has approximately 52% of the established medicinal plants that exist on earth. This makes the island particularly welcoming for rigorous scientific research on the medicinal value of plants and the development of phytomedicine thereof. Viral infections caused by the human immunodeficiency virus types 1 and 2 (HIV-1 and HIV-2), hepatitis virus B and C, influenza A virus, and the severe acute respiratory syndrome coronavirus 2 (SARS CoV-2) present a significant global burden. This is a review of some important Jamaican medicinal plants, with particular reference to their antiviral activity.

## 1. Introduction

Viral infections like the human immunodeficiency viruses types 1 and 2 (HIV-1 and HIV-2), herpes simplex virus (HSV), respiratory viruses like; rotaviruses, respiratory syncytial viruses (RSV), influenza A virus, the severe acute respiratory syndrome coronavirus 2 (SARS CoV-2) that causes the human coronavirus (COVID19), tuberculosis, and hepatitis B and C viruses, malaria, yellow virus fever (YVF), human papilloma virus (HPV), dengue-virus type 2 (DENV-2) and coxsackie virus all represent a significant global burden. In modern Western medicine, natural preventative and therapeutic alternatives are increasingly gaining attention because of greater accessibility to medicinal plants, and the possibility that they may have fewer and less adverse side-effects, are safer, and have potentially greater therapeutic efficacy than synthetic drugs. These make natural alternatives more desirable as novel drug therapies. Interest in phytoantivirals has also been renewed because of the increasing global burden of viral infections, particularly those that are newly emerging, and increasing antiviral resistance among viral strains against conventional synthetic antivirals drugs like Acyclovir and Ganciclovir.

In many parts of the world, particularly in rural, developing countries, herbalism is the only form of traditional medicine. In 2011, the WHO estimated that between 70 and 95% of the world population use botanical preparations as medicine [1]. Thousands of years of anecdotal evidence support the medicinal claims of these plants, but for the vast majority, rigorous scientific monitoring and research are required to study the mechanisms of action of the bioactive compounds, and their safety and efficacy [2]. It is estimated that some 25% of medicines on the global market are synthesized from natural products [3]. Plants from which popularly used drugs worldwide have been derived include *Ephedra sinica* Stapf. used to make methamphetamine and pseudoephedrine, Willow bark to make Aspirin, *Cannabis sativa* L. to make Dronabinol^®^, Nabilone^®^ and Sativex^®^, *Papaver somniferum* L. (opium poppy) to make morphine, codeine and heroin, *Taxus brevifolia* Nutt. to make taxol, and *Vinca rosea* L. to make vincristine.

Through ethnobotanical screening, thousands of plants and their biologically active ingredients have been identified. Of the estimated 300,000 plants species that exist worldwide, only around 15% have been evaluated for their pharmacological activity [4]. It is estimated that two-thirds of the world’s plant species have medicinal value [5]. Between 1981 and 2019, the FDA approved more than 1500 drugs either made from unaltered natural products, botanical drug mixtures, derivates of natural products, synthetic drugs with pharmacophores from natural products or mimics of natural products [6,7]. In 2019, the FDA approved 441 small molecule drugs made directly from natural products and their derivates [7]. Refer to Table 1.

Medicinal plants produce primary and secondary metabolites, that, in addition to providing health benefits to humans, may have an original intended use in the plant as biological defenses against herbivores [8]. The plant kingdom (including microbes, lichens, algae, and higher plants) produces an estimated 600,000 to 700,000 phytochemicals, with at least 150,000 to 200,000 being bioactive compounds [9].

Some active compounds in these plants responsible for the therapeutic effects of phytoantivirals include alkaloids, anthraquinones, coumarins, polyphenols (e.g., flavonoids), phenolic acids and their derivatives, lignans, naphthoquinones, peptides, nitrogenated compounds, polysaccharides and terpenes. Lipophilic terpenoids (essentials oils) disrupt the virus envelope’s lipid double layer and may even cause it to lyse [10]. Polyphenols (e.g., flavonoids) and phenolic acids may prevent the virus from docking to the host cell [10]. Phytochemicals like alkaloids (e.g., quinoline), furanocoumarins, aristolochic acid, macrozamin and cyasin can attack and mutilate nucleic acids (DNA and RNA) either by alkylating the DNA molecule or intercalating within the DNA molecule [10]. Other active ingredients that have been used in ancient/traditional medicine include strychnine, quinine, and morphine (European medicine), camptothecin and taxol (Chinese medicine), and Vinca alkaloids vincristine and vinblastine extracts from the Madagascar periwinkle (*Catharanthus roseus*) [11]. Other important biologically active ingredients that have been discovered from plants include resveratrol and curcumin.

Despite the increasing clinical research on phytoantivirals, there is need for more rigorous screening of more plants to identify new phytochemicals. Further research is required to determine the efficacy, dosage standards, optimum extraction methods/solvents, cytotoxicity/hepatoxicity, pharmacokinetics, molecular mechanisms of action, phytoantiviral screening methods, and drug interactions for many phytoantivirals.

Modern research around the world needs to now focus on the pharmacological activities of the constituent compounds of these plants and mapping their genomes and transcriptomes to produce target drugs. These compounds include alkaloids, anthraquinones, coumarins, polyphenols (flavonoids, tannins and rosmarinic acid), phenolic acids, lignans, naphthoquinones, peptides, polysaccharides and terpenes.

The top five largest orders of medicinal plants include Lamiales, Rosales, Malpighiales, Fabales and Sapindales [12]. The Medicinal Plant Database is a comprehensive multi-omics database of medicinal plants [12]. Using sequencing technology and synthetic biology, this genomic and transcriptomic mapping database will allow for the development of synthetic drugs that will be able to target a certain molecular pathway in a given disease process.

The purpose of this article is to review the state-of-the-art developments in phytomedicines in Jamaica, with particular focus on antivirals. This should provide a valuable reference for further studies on development and clinical translation of these phytomedicines.

## 2. Jamaican Plants with Medicinal Potential

Medicinally, there is increasing focus on Jamaica because of the wide diversity of medicinal plants. Plant-based decoctions have been a significant part of Jamaican traditional folklore medicine primarily to treat the common cold, flu, headache, nausea, pain, reproductive system disorders and digestive issues, among some of the aforementioned diseases that are a significant global burden. Popular medicinal plants used in traditional Jamaican folk medicine include *Momordica charantia* L. (Cerasee), *Aloe barbadensis* Miller (*Aloe vera* (L.) Burm.f./), *Cannabis sativa* L. (Ganja)*, Cola acuminate* (P.Beauv.) Schott and Endl. (*Bissy), Morinda citrifolia* L. (Noni), *Pothomorphe* umbellata (L.) Miq. (Cowfoot Leaf), *Cinnamomum tamala* (Buch.-Ham.) T.Nees and Eberm. (Bay leaf)*, Zingiber officinale* Roscoe (Ginger), *Bryophyllum pinnatum* (Lam.) Oken (Leaf-of-life), *Moringa oleifera* Lam. (Moringa), *Panax ginseng* C.A.Mey. (Ginseng), *Mikania micrantha* Kunth (Quako), *Marrubium Vulgare* L. (*White horehound/”Mint”), Andrographis paniculata* (Burm.f.) Nees (Rice bitters)*, Curcuma Longa* L. (turmeric), *Petiveria alliacea* L. (Guinea Hen Weed), *Camellia Sinensis* (L.) Kuntze (*“Tee tree”), Alysicarpus vagilinis* (L.) DC. (Medina) and *Allium sativum* L. (Garlic). Jamaicans have used a combination of these plants for detoxing and to treat a wide range of ailments including, but not limited to headache, nausea, pain, inflammation, cancer, diabetes, reproductive disorders, viral infections (common cold and “flu”), arthritis and hypertension.

The antiviral properties of these medicinal plants are supported by traditional anecdotal evidence. There is a need for more scientific research to establish efficacy, pharmacological and pharmacokinetic data and standard dosages for medical conditions. Viral infections are the second-most identified health condition for which medicinal plants were used to treat in Jamaica [13].

Popular in the Jamaican ethnomedical heritage is what is known as a “*root tonic*” or “*strong back*”. This drink is a decoction of plants including, but not limited to *Smilax balbisiana* Kunth (**“chany root”), *Smilax ornata* (sarsaparilla), *Zingiber officinale* Roscoe (“ginger*”), Alysicarpus vaginalis* (L.) DC. (“medina”), *Morinda royoc* L. (“redgal”), *Desmodium incanum* DC. (“tick clover”), *Cuphea parsonsia* (L.) R.Br. ex Steud. (commonly referred to as “strong back leaf*”), Trophis racemosa* (L.) Urb. (“ramoon”), and *Iresine diffusa* Humb. and Bonpl. Ex Willd (“nerve west”) [14]. This decoction is commonly used by males to treat impotence, and increase stamina. However, many of these plants like *Z. officinale*, *C. parsonsia, S. ornata*, and *A. vaginalis* have noted antiviral activity.

Table 2 shows the frequently used medicinal plants to treat the common cold and flu in Jamaica.

A 2011 ethnomedicinal survey by Picking and colleagues investigated popular medicinal plants in Jamaica and confirmed the significance of plants and herbs in primary health care in Jamaica [13]. This survey followed the TRAMIL network ethnomedicine methodology. The TRAMIL network is a Caribbean-based network of collaborators conducting scientific evaluations of medicinal plants in the region. The questionnaire was administered to 407 randomly selected adults, from randomly selected geographical clusters across Jamaica. The survey revealed that respondents used 116 medicinal plants for various ailments. Of these, 94% (107 plants) were distributed across fifty-one plant families. The top five families with the most frequent plant families identified were *Fabaceae*, *Lamiaceae*, *Asteraceae*, *Malvaceae*, and *Piperacerae* [13]. Common plants of the *Fabaceae* family include Legumes, Maranga, Strong Back, Dandelion and Medina. Common herbs of the *Lamiaceae* family include Basil, Sage, Rosemary, Oregano, Thyme, Mentha and Lavender. The *Lamiaceae* family is commonly known as the Mint family. Some plants of the *Asteraceae* family include of Marigold, Spanish Needle, and Quaco Bush. Common plants of the *Malvaceae* family include Bissy, Sorrel, and Hibiscus.

Of the 107 plants identified by survey respondents, eight are endemic to Jamaica. These are *Piper amalago* L. (Pepper elder), *Rhytidophyllum tomentosum* (L.) Mart. (“Search-mi-heart”), *Bidens reptans* (L.) G.Don (“McKatty Weed”/”Marigold”), *Peperomia amplexicaulis* (Sw.) A. Dietr (“Jackie’s saddle”), *Oryctanthus occidentalis* (L.) Eichler (“Godbush”), *Pilea microphylla* (L.) Liebm. (“Baby puzzle”), *Smilax balbisiana* Kunth (“Chany Root”), and *Boehmeria jamaicaensis* Urb. 1907 (“Doctor Johnson”). Of these, *P. amalago* L. has the most notable medicinal use [13].

The common cold, which may be caused by different viruses, has been among the most prevalent health issues in Jamaica. Leaf of Life (*B. pinnatum*) and Jack-in-the bush (*Euphatorium odoratum* L.) have a frequency of use of equal to or greater than 20% for these viral infections in Jamaica [13].

A 2006 study by Mitchell and Ahmad investigated and summarized all the medicinal plant research carried out on Jamaican medicinal plants between 1948 and 2001 at the Faculty of Pure and Applied Science, University of the West Indies (UWI), Mona, Jamaica [15]. Mitchell and Ahmad report that Jamaica has at least 334 medicinal plant species. However, only 193 were tested for their bioactivity—80 plants of which were reported to have reasonable bioactivity. Natural products were also identified from 44 of these plants. Of the plants tested at UWI, only 31 were endemic to Jamaica. Some 23% percent of these 31 plants were tested and found to be bioactive [15].

## 3. Jamaican Plants with Major Antiviral Activity

Of the many plants which have important antiviral activity, those used most frequently in Jamaica for the common cold and influenza viruses are: Cerassee (*Momordic charactia* L.), and Fevergrass (*Cymbopogon citratus* L.), Tamarind leaves (*Tamarindus indica* L.)*,* Thyme (*Thymus vulgaris* L.)*,* Sarsaparilla (*Smilax ornata* Lem.), Soursop leaf (*Annona muricate* L.)*,* Pimento Leaf (*Pimenta dioca* (L.) Merr.), Garlic (*Allium sativum* L.), Leaf of Life (*Bryophyllum pinnatum* (Lam.) Oken), Search-mi-heart (*Rhytidophyllum tomentosum (L. Mart.*), Mint (*Mentha piperita* L.), Cinnamon (*Cinnamomum verum* J. Presl), Oregano (*Oregano vulgare* L.), Papaya leaf (*Carica papaya* L.), and Medina (*Alysicarpus vaginalis* (L.) DC.). Some Jamaican plants with major antiviral activity are shown in Figure 1 and are discussed below.

### 3.1. Ball Moss/ “Old Man’s Beard” (Tillandsia recurvata L.)

The *Tillandsia* L. genus is made up of approximately 650 species and belongs to the Bromeliaceae family endemic to North and South America and the Caribbean. Pineapples, although of a different genus (*Ananas*), are also of the Bromeliaceae family. These species differ in growth habit, trichrome distribution, type of fruit, seed, leaflet, photosynthesis, type of pollinator, and its epiphytes [16]. *T. recurvata* has been used in traditional medicine to treat kidney inflammation (Bolivia), rheumatism, ulcers and hemorrhoids (Brazil), menstrual disorders (Mexico), eye infection (Uruguay), and Leucorrhea (USA). Most species of the *Tillandsia* L. genus are epiphytes on trees, inert substrates, rocky slopes, phone wires, electricity lines, or may survive independently in soil. It should be noted that *Tillandsia usneoides* (L.) L. (“Spanish Moss”) and *T. recurvata* (“Ball Moss”) are not mosses but, instead, are angiosperms—produces flowers. The primary compounds in Tillandsia genus are triterpenoids and sterols (51%), flavonoids (45%) and cinnamic acid (4%). Lowe and colleagues discovered Dicinnamoyl-Glycerol Esters in the ball moss plant, with both anti-cancer and anti-HIV properties (US patent #US8907117B2) [17]. These anti-cancer and antiviral properties may also be due to the naturally occurring cycloartanes found in the ball moss plant—cycloartane-3,24,25-diol and cycloartane-3,24,25-triol [18]. A 2020 study by Gao and colleagues also confirmed anti-HIV properties of tillandsia [19]. Figure 2 below are examples of chemical structures of some bioactive molecules found in *Tillandsia recurvata* L.

### 3.2. Aloe vera (L.) Burm. f. (Aloe barbadensis Miller)

The antiviral activity of *A. vera* may be due to a number of phytochemicals including vitamins, minerals, anthraquinones, polyphenols (e.g., flavonoids), polysaccharides, phenolic acids and sterols [20]. A 2018 study by Gansukh and colleagues investigated the anti-influenza activity of flavonoids and phenolics found in *A. vera* [21]. A 5 min ultrasonic water extraction of aloe emodin, an anthraquinone prepared from aloin from the *A. vera* plant, showed anti-influenza activity with zero cytotoxicity [21]. Aloe emodin was also able to inactivate herpes simplex virus types 1 and 2, varicella-zoster virus, pseudorabies virus, and the influenza virus, by partial destruction of the virus’ envelope [22]. Lectins, extracted from the gel portion of the leaves of *A. vera,* also showed antiviral activity against the human cytomegalovirus (CMV) in cell culture [23].

*A. vera* anti-HSV-1 activity was also assessed and confirmed in a 2016 study by Rezazadeh and colleagues [24]. A preparation of *A. vera* gel showed significant (0.2-5%) inhibition of herpes simplex virus-1 cells isolated from the lip lesions of a patient and grown cell culture (Vero cells) [24]. *A. vera* was also shown to inhibit HSV-2 attachment and post-attachment processes, and entry into Vero cells [25]. Figure 3 below are examples of chemical structures of some bioactive molecules found in *A. vera.*

### 3.3. Ganja (Cannabis sativa L.)

It is estimated that *C. sativa* produces an estimated 545 [26] chemical compounds belonging to biogenetic classes [27]. Of these compounds, the two most prominent and most studied are the secondary metabolites (phytocannabinoids)—Δ^9^-tetrahydrocannabinol (Δ^9^-THC) and cannabidiol (CBD), both of which have a wide therapeutic window against many ailments. In addition to these secondary metabolites, *C. sativa* produces hundreds of non-cannabinoids secondary metabolites, also with a wide range of therapeutic applicability against many ailments. These include terpenoids, flavonoids, stilbenes, lignans, and alkaloids [28]. Some of these phytochemicals have antiviral activity. Dihydro-resveratrol, a metabolite of trans-resveratrol, an antiviral found in grapes, is also found in cannabis [27]. Terpenes like limonene and ocimene have also been reported to demonstrate antiviral activity [29,30]. A 2020 study by Ngwa and colleagues reported that a small antiviral flavonoid molecule Caflanone has selective activity against the human coronavirus hCov-OC43 (COVID-19) disease [31].

Cannabidiol (CBD) is one of 100 pharmacologically active terpenophenic/lipophilic compounds called Cannabinoids—found in *C. sativa*. CBD It is the major non-psychoactive/non-intoxicating component of the cannabis plant. This simply means that it does not get you “high”. CBD still, however, retains its therapeutic properties and benefits.

CBD can be used to regulate the immune system’s response to viruses (and other invading pathogens. The immune system uses oxidative stress (via reactive oxygen species) to combat invading pathogens. During many disease processes, the cells of the body accumulate high levels of reactive oxygen species (R.O.S.). This may be due to reasons including, but not limited to increased metabolic activity, increased oxidase activity, and/or increased mitochondrial activity. In excess, oxidative stress can cause tissue and organ damage. Numerous studies show CBD’s potential as an anti-oxidant. CBD inhibits neurotoxicity and oxidative stress by reducing inflammation, production of reactive oxygen species and other oxidative stress parameters [32] associated with infections by pathogens. A proposed mechanism of action of CBD is the amelioration of ROS production. When oxygen is metabolized in the body, ROS is produced. In moderation, ROS is involved in homeostasis and signaling and is associated with the maintenance of healthy cells [32]. Antioxidants like CBD mitigate excessively produced reactive oxygen species within the cells, and in doing so, reduce the oxidative stress in cells [5]. Antioxidants may therefore have therapeutic applicability against many human diseases including but not limited to viral infections, but also cancer cardiovascular diseases and inflammatory diseases [5]. Figure 4 below are examples of chemical structures of some bioactive molecules found in *C. sativa* L.

#### 3.3.1. Cannabidiol (CBD) and HIV/AIDS

CBD is used to alleviate the wasting syndrome associated with HIV and AIDS [33]. It is used as an antiemetic and orexigenic agent (appetite stimulant), and may generally just improve the overall quality of life of an HIV/AIDS patient. Anecdotal evidence suggests that CBD in HIV/AIDS patients may improve appetite, reduce nausea and vomiting, increase caloric intake, promote weight gain, improve memory and dexterity, improve mood, and mitigate the negative side effects of current anti-retroviral therapeutic agents [33]. In terms of disease progression (morbidity) and delaying the likelihood of death from HIV/AIDS, current studies show that CBD is not effective [33].

#### 3.3.2. Cannabidiol (CBD) and Hepatitis Viruses

Liver disease in general, is a major global health burden. Viral hepatitis is a disease of the liver characterized by liver inflammation and damage as a result of viral infection. Viral hepatitis is commonly caused by one of five hepatotropic viruses (hepatitis A, B, C, D and E), but may be caused by other viruses like the herpes simplex virus (HSV) Yellow fever virus (YFV), cytomegalovirus (CMV) and Epstein–Barr virus (EBV). Hep A, Hep B, and Hep C are the most common causes of viral hepatitis. Hep A and Hep E are spread by the fecal-oral route, that is, contamination via contaminated food or water. Hep B, Hep C and Hep D are spread through blood transfusion. There is evidence that these may also be spread sexually.

Hepatitis may also be caused by other types of micro-organisms including bacteria, fungi and even parasites, non-infectious agents like drugs and alcohol, and other metabolic and autoimmune diseases [34]. Hepatitis infections may either be acute (short-term), where the body will be able to resolve the infection or chronic (long-term), where the body is unable to resolve the infection, resulting in liver failure, liver cirrhosis and liver cancer.

In a 2017 in vitro study by Lowe and colleagues explored the bioactivity of CBD against hepatitis B and C viruses [35]. The anti-hepatitis B assay was carried out using HepG2 2.2.15 cells that produce high levels of the HBV wild-type ayw1 strain. The anti-hepatitis C assay utilized Huh7.5 cells mixed with cell-culture derived HVC. In both assays, cells were plated in a 96-well microtiter plate. A single concentration of 10μM of the test compound (CBD) was then added to both microtiter well plates, incubated, and an analysis of antiviral activity was determined by calculating the percent of inhibition of viral replication. CBD was shown to have inhibitory effects against viral hepatitis C (HBC) but not viral Hepatitis B (HBV). In a dose–response assay, at a single concentration of 10 µm, CBD was able to dose-dependently inhibit HCV replication by 86.4% [35]. CBD also seems to have therapeutic efficacy against autoimmune/non-viral hepatitis [35]. CBD shows in vivo activity through its interaction with the CB2 receptor. This interaction inhibits the pathogenesis of autoimmune hepatitis by inducing the apoptosis of thymocytes and splenocytes. This in turn, inhibits T-cells and macrophages attacking the liver thereby inhibiting the release of pro-inflammatory cytokines [35].

Myeloid-derived suppressor cells (MDSCs) are responsible for regulating the immune system by suppressing T-cell function and inhibiting liver inflammation. Through interaction TRPV1 receptor, CBD is shown to activate MDSCs, thereby inhibiting inflammation and hepatitis in a murine model [36]. In a concanavalin A model of acute hepatitis in mice, Hegde and colleagues report that CBD was able to reduce ConA-induced inflammation by inhibiting the production and release of various pro-inflammatory cytokines, and protect the mice from acute liver injury [36].

#### 3.3.3. Caflanone (a Non-Cannabinoid Secondary Metabolite Found in *C. sativa* L.)

Caflanone is a small phytoantiviral flavonoid molecule with selective activity against the human coronavirus hCov-OC43 (COVID-19) disease belonging to clade b of the genus Betacoronavirus same as SARS-COV-2 [34]. In preclinical studies, Caflanone inhibited the hCov-OC43 human Coronavirus with an EC50 of 0.42 µM [31]. In silico studies show that the Caflanone molecule carries out its prophylactic mechanism of action by inhibiting the Angiotensin-converting enzyme 2 (ACE2) receptor found in the lung and respiratory tract, used by the virus to cause an infection [31]. Caflanone was also shown to have strong binding affinity to two of the proteases (PLpro and 3CLpro) are vital to the replication of SARS-COV-2 in humans, which would inhibit viral entry to and/or replication within human cells [31].

The following phytoantivirals were investigated and compared to Chloroquine (CLQ), a potential COVID-19 prophylactic and therapeutic agent currently in clinical trials. The docking/binding studies results below show that the phytoantiviral flavonoids (Hesperetin, Myricetin, Linebacker, and Caflanone) could bind equally or more effectively than CLQ [31].

#### 3.3.4. Terpenoids of Cannabis *Sativa* L. as Antiviral Agents

The Medical Cannabis Network of Israel also reports that a current study is being undertaken by researchers at the Israel Institute of Technology investigating the therapeutic efficacy of a cannabis terpene inhalant formulation in suppressing the immune system response against COVID-19 [37]. A molecular docking analysis also reported the antiviral activity of Ginkgolide A, a terpenoid produced by the *Ginkgo biloba* tree, against COVID-19 [38].

## 4. Guinea Hen Weed (*Petiveria alliacea* L.)

*P. alliacea* is a herbaceous shrub belonging to *Petiveriaceae*, the pigeonberry family. It is native to tropical regions like Africa, India, the Caribbean, tropical areas of North and Central America. *P. alliacea* produces a number of bioactive compounds including polyphenols, alkaloids, tannins, coumarins, steroids, essential oils, dibenzyl trisulfide, and flavonoids [39]. These phytochemicals are responsible for Guinea Hen Weed’s wide therapeutic window as an anxiolytic, anticonvulsant, antinociceptive, neuroprotector, cognitive enhancer, and antidepressant [39].

Dibenzyl trisulfide/DTS (C_14_H_14_S_3_) in the Guinea Hen Weed is responsible for its anti-cancer, ant-HIV, and anti-hepatitis C properties [40,41]. Figure 5 below should the chemical structure of dibenzyl trisulfide (DTS) found in Guinea Hen Weed.

The anti-cancer and antiviral properties of this compound may be due to its interaction with the mitogen-activated protein, extracellular-regulated kinases 1 and 2 (MAP Kinases Erk 1/Erk2) pathway [41] (Figure 6). The mechanism by which DTS inhibits HIV-1 activity is by inhibition of the reverse transcriptase (RT) activity of the HIV-1 virus [40] (Figure 7). Anecdotal evidence also suggests that Guinea Hen Weed may also be used to combat the influenza viruses. DTS has also been reported to demonstrate anti-hepatitis C virus properties responsible for hepatocellular carcinoma [42]. However, the molecular mechanism of action has not yet been elucidated. 

## 5. Ginger (*Zingiber Officinale* Roscoe)

Ginger is frequently used in Jamaica as an antiemetic and antinauseant. In addition to having antioxidant, antimicrobial, anti-inflammatory, and anticoagulant properties, fresh ginger, as opposed to dried, has also been reported to be antiviral, inhibiting human respiratory syncytial virus-induced plaques in human upper (HEp-2) and low (A549) respiratory tract cell lines [46]. 

The bioactive compounds in ginger that are responsible for the wide therapeutic window of ginger include gingerols (like shogaols, 6-, 8- and 10- gingerol), paradols, ketones, phenolic compounds (like 6-dehydrogingerdione, zingerone, quercetin, and gingerenone-A), terpenes (like α-curcumene, α-farnesene, β-bisabolene and β-sesquiphellandrene), esters, aldehydes, and alcohols [47]. Figure 8a–e are examples of bioactive molecules found in *Zingiber officinale* Roscoe and their chemical structures.

A possible mechanism of antiviral action of ginger is the blocking of viral attachment and internalization via stimulation of mucosal cells of the respiratory tract to secrete interferon- β (IFN- β) [46] (Figure 9). Ginger has also been reported to demonstrate antiviral properties against the hepatitis C virus (HCV) [47].

## 6. Turmeric (*Curcuma Longa* L.)

Turmeric (*Curcuma longa* L.) is the belowground portion (rhizome) of the ginger plant, of the family Zingiberaceae. Turmeric is a widely used spice, food preservative, food coloring and dye but has also been employed for its medicinal value [48]. It has a wide therapeutic window as an antiviral, aseptic, antibacterial, antifungal, anti-cancer, anti-phlegmatic antiprotozoal, antioxidant, anti-inflammatory, antifibrotic, antifertility, antiulcer, antivenom, anticarcinogenic and anticoagulant [48]. In traditional Eastern medicine it is used to treat respiratory ailments, eating disorders, digestive disorders, hepatic disorders and biliary disorders [48].

The primary phenolic compounds (curcuminoids) produced by turmeric are curcumin (diferuloylmethane), demothoxycurcumin and bisdemethoxycurcumin [48]. Other bioactive compounds produced include sodium curcuminate, essential oils (e.g., turmerone, zingiberene and sesquiterpines), monoterpenoids, and sesquiterpenoids. In addition to being an antiviral, curcumin is anti-inflammatory, anti-tumorigenic and anti-oxidant [49]. It is also responsible for the orange–yellow color of turmeric. Figure 10 displays the chemical structure of curcumin.

An in vitro study reported the antiviral activity of curcumin against three major molecular pathways responsible for Epstein–Barr virus (EBV) reactivation, and was shown to inhibit the transcription of BamH fragment Z left frame 1 (BZLF1) gene [50] which plays a significant role in lytic EBV DNA replication. The anti-HIV-1 activity of curcumin was also assessed and confirmed via inhibition of the HIV-1 integrase [51] (Figure 11). Turmeric was also shown to display slight antiviral activity against respiratory viruses.

A 2007 study by Huang and colleagues demonstrated activity against the influenza viruses, parainfluenza viruses 1, 2 and 3, adenovirus, and respiratory syncytial virus [52]. Curcumin was also reported to inhibit H1N1, H6N6, herpes virus, coxsackievirus B3 (CVB3), human T-cell leukemia virus type 1, hepatitis C virus, high risk human papillomaviruses 16 and 18 (HPV-16 and HPV-18) [53].

## 7. Moringa (*Moringa oleifera* Lam.)

*M. oleifera* is popular tree native to India where it is widely eaten as a vegetable and utilized in traditional medicine. It is also used in the Western hemisphere for the same purposes. Moringa has a wide window of benefits, due primarily to bioactive compounds like phenolic compounds, saponins, tannins, amino acids, proteins, phytates, tocopherols (γ and α), carbohydrates, unsaturated fatty acids, oils, antioxidant compounds, and glucosinolates [54]. Other bioactive compounds include vitamins A, B1, B2, B3, B7, C, D, E, K, calcium, potassium, iron, magnesium, phosphorus and zinc [55]. The pharmacological compounds found in the moringa plant have medicinal properties including antimicrobial activity against viruses, bacterial, fungi and parasites. It is also generally known to boost the immune system.

Leaf extracts from Moringa oleifera showed anti-herpetic activity and were reported to successfully inhibit the growth of HSV-1 and 2 in Vero cells by 43.2 and 21.4%, respectively [56]. Ethanolic leaf extracts of Moringa oleifera were also reported to have anti-influenza activity in vitro [57]. A 1998 study by Murakami and colleagues reported that niaziminin, a thiocarbamate, 4-[(4′-O-acetyl-alpha-L-rhamnosyloxy)benzyl] isothyanate (ITC) and allyl- and benzyl-ITC (two isothyanate-related compounds), were all able to inhibit Epstein–Barr virus (EBV) in vitro [58]. Figure 12 shows the chemical structures of niaziminin, allyl-ICT, and benzyl-ICT.

In another in vitro study, the effects of an aqueous extract of *M. oleifera* (leaves) were able to decrease the expression of hepatitis B virus genotypes C and H in Huh7 cells [59].

## 8. Lignum Vitae (*Guaiacum officinale* L.)

The *Lignum vitae* (which translates to “wood of life”) is native to the Caribbean and South America. The wood has many industrial purposes, for example to make furniture. The flower is the national flower of Jamaica. In tradition Caribbean medicine, it is used as a stimulant, antiseptic, and to treat syphilis [60]. In Trinidad and Tobago, it is used to control fertility [60]. A 2014 study by Lowe and colleagues screened and investigated the anti-HIV properties of leaf, seed and twig extracts of the lignum vitae plant biomass against HIV-1 (strain HIV-1 BaL) infected peripheral blood mononuclear cells (PBMCs) [61]. The main pharmacological compounds in this plant are saponins, including guaianin A, guaianin B and guaianin N [61]. All the types of extracts/compounds tested inhibited HIV-1 replication in the infected PBM cells. Figure 13 shows chemical structures of some bioactive molecules found in lignum vitae.

## 9. Garlic (*Allium sativum* L.)

*A. sativum* is a member of the onion genus, *Allium*. It is another spice that is traditionally used in fundamental dishes around the world, but has also been used in traditional medicine for thousands of years. It has a wide therapeutic window as an antiviral, antioxidant and antibacterial against both Gram-negative and Gram-positive bacteria, antifungal activity against *Candida albicans*, and is antiprotozoal against *Giardia lamblia* and *Entamoeba histolytica* [62]. Anecdotal and scientific data also confirm garlic’s therapeutic efficacy against diabetes, cancer, free-radical damage, atherosclerosis, heavy metal intoxication and hyperlipidemia. Garlic produces many bioactive compounds including flavonoids, enzymes, fructo-oligosaccharides, Maillard reaction products, and organosulfur compounds like S-allylcysteine, diallyl polysulfides, alliin, vinyldithins, allicin (diallyl thio-sulfinate), allyl methyl thiosulfinate, and ajoene [63]. These are responsible for garlic’s wide therapeutic window, including as an antiviral against *influenza A* and *B, HSV-1* and *HSV-2*, common cold, and viral pneumonia among other respiratory viruses [64]. Figure 14 is a graphical representation of the conversion of organosulfur compounds in garlic. Organosulfur compounds demonstrate antiviral activity via inhibition of various stages of the general virus life cycle including viral attachment, entry and multiplication [64] (Figure 15). Figure 16a–d show the chemical structures of some bioactive molecules found in garlic.

These compounds are released and may be extracted from the cells of fresh, crushed garlic bulbs, and are responsible for the characteristic garlic smell. This is why it is usually eaten raw. Allicin is the main bioactive compound found in garlic [62]. The bioactivity of allicin and other organosulfur compounds may be attributed to their chemical interaction with thiol groups of other molecules [65].

A 2009 study by Mehrbod and colleagues reported garlic’s antiviral activity against the influenza virus in vitro [65], further confirming the anecdotal evidence for its frequency of use in Jamaica against the flu and to “boost the immune system”. Garlic is frequently used in Jamaica to treat the common cold, too. A 2001 study by Josling evaluated an allicin-containing supplement against the common cold and confirmed this bioactivity [66]. In another study, an aged garlic extract supplement was reported to boost the immune system by inducing NK cell function and proliferation of γδ-T cells [67]. In vitro studies of garlic extracts were also shown to produce anti-influenza B and anti-herpetic activity (anti-*HSV-1*) [68]. This bioactivity was dose-dependent.

Further studies are required to assess the antiviral activity of garlic in human and animals. Garlic is also reported to be good for the cardiovascular system, having the ability to lower systolic blood pressure and ultimately treat uncontrolled hypertension [69]. A 2013 study by Ashraf and colleagues also confirm garlic’s dose-dependent and duration-dependent ability to significantly lower both systolic blood pressure and diastolic blood pressure [70]. Garlic is further reported to be good for lowering cholesterol and thus may be used to treat hypercholesterolemia (a form of hyperlipidemia) [71].

## 10. Sorrel (Hibiscus sabdariffa L.)

In Jamaica the fresh calyx of the flower of sorrel is commonly made into a drink and eaten in salads. However, it possesses significant medicinal value and may have applicability as a diuretic, sedative, emollient, demulcent antiscorbutic, analgesic, purgative, antipyretic, cholagogue, antiseptic, anti-tumorigenic, anti-cancer, aphrodisiac, and may be used to treat dyspepsia, disorders of the heart, hypertension, biliary disorders and abscesses [72]. Sorrel is also known for antimicrobial, antioxidant, and anti-inflammatory properties. The antiviral activity of sorrel was also assessed and shown to inhibit HSV-2 in vitro with safe cytotoxicity [73]. Sorrel was also reported to have anti-human influenza A virus activity in vitro [74]. This bioactivity attributed to sorrel may be attributed to a number of secondary metabolites including saponins, flavonoids (e.g., quercetin, gossypitrin and hibiscetin-3-glucoside, organic acids (e.g., protocatechuic, malic acid, hydroxycitric acid, and hibiscus acid), anthocyanins (e.g., anthocyanidin and delphinidin-3-O-sambubioside), glycosides, alkaloids, phenolic acids and phenolic compounds (e.g., α-tocopherol) [75,76]. Some of these secondary metabolites and their chemical structures are shown in Figure 17 below.

**Table 2 molecules-26-00607-t002:** Summary of frequently used medicinal plants in Jamaica.

	Plant	Origin	Anti-Viral Window	Part of Plant Used to Make Preparation	Method of Preparation/Admin	Proposed Antiviral Mechanism (s)
1.	Ginger (*Zingiber officinale* Roscoe)	Native to Asia. Cultivated widely in Jamaica.	Anti-viral activity against the respiratory syncytial virus (RSV) [46], hepatitis C virus (HCV) [47], common cold (anecdotal), and the human influenza viruses (anecdotal).	Root Nodule/Rhizome	Decoction made from 1 tsp or ~1 inch of fresh root nodule brewed in 1 cup of water. Steep for ~10 min. Sweeten as desired. This is to treat the common cold and flu.	Inhibition of viral attachment and internalization via stimulation of mucosal cells of the respiratory tract to secrete interferon-β (IFN-β) [46].
2.	*Turmeric* (*Curcuma longa* L.)	Cultivated (Not indigenous to Jamaica)	Anti-viral activity againsthepatitis C virus (HCV), Epstein–Barr virus (EBV) [50], HIV-1 [51], human influenza viruses- H1N1, H6N6 [53], parainfluenza viruses 1, 2 and 3 [53], vesicular stomatitis virus (VSV) [53] and respiratory syncytial virus [53].	Root Nodule/Rhizome	Decoction made from 1 tsp or ~1 inch of fresh root nodule brewed in 1 cup of water. Steep for ~10 min. Sweeten as desired. This is to treat the common cold and flu.	Inhibition of Epstein–Barr virus BZLF1 transcription in Raji DR-LUC cells [50]. Inhibition of HIV-1 integrase [51].
3.	Ball Moss/ “Old Man’s Beard” (*Tillandsia recurvata* L.)	Indigenous to Jamaica	Anti-viral activity against HIV [17,18,19], common cold (anecdotal),and flu (anecdotal)	Leaf (Fresh or dried)	Decoction made from 1 tsp or ~1 inch of fresh root nodule brewed in 1 cup of water. Steep for ~10 min. Sweeten as desired. This is to treat the common cold and flu.	Mechanism of antiviral activity unknown.
4.	*A. vera* (*Aloe barbadensis* Miller)	Cultivated (Not indigenous to Jamaica)	Anti-viral activity against the human influenza viruses [21,22], varicella-zoster virus (VZV) [22], cytomegalovirus (CMV) [22], pseudorabies virus [22], herpes simplex virus types 1 and 2 (HSV-1 and -2) [24,25], and common cold (anecdotal)	Inner-leaf, gel-like pulp.	Juice made of inner gel-like pulp. This is to treat the common cold and flu.	Partial destruction of the virus’ envelope (herpes simplex virus types 1 and 2, varicella-zoster virus, pseudorabies virus, and the influenza virus) [22]. Inhibition of HSV-2 attachment and post-attachment processes, and entry into Vero cells [25].
5.	*Ganja* (*Cannabis sativa* L.)	Cultivated (Not indigenous to Jamaica. Indigenous to Central Asia)	Anti-viral activity against HIV/AIDS wasting syndrome [33], hepatitis viruses [35], human coronavirus [34], and common cold (anecdotal).	Leaf (Fresh or dried)	Decoction made of 1–2 tsp of dried leaves brewed in 1 cup of water. Steep for ~10 min. Sweeten as desired. This is to treat the common cold and flu.	Cannabidiol (CBD) inhibits replication of hepatitis B and C viruses [35]. CBD also inhibits the pathogenesis of autoimmune hepatitis by inducing the apoptosis of thymocytes and splenocytes [35]. Caflanone inhibits the human coronavirus (hCov-OC43) via inhibition of the Angiotensin-converting enzyme 2 (ACE2) receptor found in the lung and respiratory tract [31].
6.	Guinea Hen Weed (*Petiveria alliacea* L.)	Cultivated (Not indigenous to Jamaica. Native to Central and South America)	Anti-viral activity against HIV [40], common cold (anecdotal), human influenza viruses (anecdotal), and hepatitis C virus (HCV) [42].	Whole plant (roots stem, and leaves (fresh or dried)	Decoction made of 1–2 tsp of dried leaves brewed in 1 cup of water. Steep for ~10 min. Sweeten as desired. This is for the common cold and flu.	DTS inhibits HIV-1 activity is by inhibition of the reverse transcript-tase (RT) activity of the HIV-1 virus [40].
7.	Moringa (*Moringa oleifera* Lam.)	Cultivated (Not indigenous to Jamaica. Native to South Asia)	Anti-viral activity against the human influenza viruses (anecdotal), HSV-1 and 2 [56], Epstein–Barr Virus (EBV) [58], and hepatitis B virus (HBV) [59].	Leaf (primarily dried)	Decoction made of 1–2 tsp of dried leaves brewed in 1 cup of water. Steep for ~10 min. Sweeten as desired. This is for the common cold and flu.	The antiviral mechanism of action is unknown.
8.	*Lignum Vitae* (*Guaiacum officinale* L.)	Cultivated (Native to the Caribbean)	Anti-viral activity against HIV [61], common cold (anecdotal), and flu (anecdotal).	Flowers, Leaves (fresh or dried), Powdered bark,	Decoction made of 1–2 tsp of dried leaves brewed in 1 cup of water. Steep for ~10 min. Sweeten as desired. This is for the common cold and flu.	The antiviral mechanism of action is unknown.
9.	Garlic (*Allium sativum* L.)	Cultivated (Native to Asia)	Anti-viral activity against the human influenza viruses (anecdotal), parainfluenza virus type 3, HSV-1 and -2 [64], and common cold (anecdotal).	- Raw bulb (most effective), Cooked, Supplement, or garlic extract.	Eat or cook raw bulb (2–3) to combat the common cold and flu viruses.	The antiviral mechanism of action is unknown.
10.	Sorrel (*Hibiscus sabdariffa* L.)	Cultivated (Not native to Jamaica)	Anti-viral activity against HSV-2 virus [73], and human influenza viruses [74], Common cold (anecdotal),	Calyx	Juice made from calyx. Calyx is left to brew overnight in boiled water, cooled and strained, then sweetened to taste.	The antiviral mechanism of action is unknown.
11.	“Search-mi-heart” (*Rhytidophyllum tomentosum* (L.) Mart.	Cultivated. Native to South America and the Caribbean	Common cold and flu (anecdotal)	Leaf (primarily dried)	Decoction made of 1–2 tsp of dried leaves brewed in 1 cup of water. Steep for ~10 min. Sweeten as desired.	The antiviral mechanism of action is unknown.
12.	Pepper elder (*Piper amalago* L.)	Cultivated (Not native to Jamaica)	Common cold and flu (anecdotal)	Fresh Leaves	Decoction made of 1–2 tsp of dried leaves brewed in 1 cup of water. Steep for ~10 min. Sweeten as desired.	The antiviral mechanism of action is unknown.
13.	McKatty Weed/Marigold (*Bidens reptans* (L.) G.Don	Cultivated (Not native to Jamaica)	Common cold and flu (anecdotal)	Flowers and/or Leaves (primarily dried)	Decoction made of 1 tbsp of dried leaves brewed in 1 cup of water. Steep for ~10 min. Sweeten as desired.	The antiviral mechanism of action is unknown.
14.	Chany Root a.k.a. Jamaican Sasparilla (*Smilax balbisiana* Kunth)	Native to Jamaica	Common cold and flu (anecdotal)	Root	Decoction or tonic made of pre-soaked roots brewed in 1 cup of water. Steep for ~10 min. Sweeten as desired.	The antiviral mechanism of action is unknown.
15.	Lemongrass/Fevergrass(*Cymbopogen citratus* L.)	Cultivated (Not native to Jamaica)	Common cold and flu (anecdotal)	Leaf (Fresh or dried)	Decoction made from a handful of fever grass leaves and stems in 1 cup of water. Steep for ~10 min. Sweeten as desired.	The antiviral mechanism of action is unknown.
16.	Cerasee (*Momordica charantia* L.)	Cultivated (Not native to Jamaica. Native to Africa and Middle East)	Common cold and flu (anecdotal)	Leaf (Fresh or dried) and stems	Decoction made from a handful of cerasee leaves and stems in 1 cup of water. Steep for ~10 min. Sweeten as desired.	The antiviral mechanism of action is unknown.
17.	Soursop (*Annona muricata* L.)	Cultivated (Not native to Jamaica. Native to South America)	Common cold and flu (anecdotal)	Leaf (Fresh or dried)	Decoction made from 2–3 leaves in 1 cup of water. Steep for ~10 min. Sweeten as desired.	The antiviral mechanism of action is unknown.
18.	Leaf of life (*Bryophyllum pinnatum* (Lam.) Oken	Cultivated (Not native to Jamaica. Native to Africa)	Common cold and flu (anecdotal)	Leaf (Fresh or dried)	Decoction made from 2–3 leaves in 1 cup of water. Steep for ~10 min. Sweeten as desired.	The antiviral mechanism of action is unknown.
19.	Dogblood (*Rivina humilis* L.)	Cultivated (Not native to Jamaica)	Common cold and flu (anecdotal)	Whole plant, but primarily leaf (Fresh or dried) and stem	Decoction made of 1–2 tsp of dried leaves brewed in 1 cup of water. Steep for ~10 min. Sweeten as desired.	The antiviral mechanism of action is unknown.
20.	Lime Leaf Tea (*Citrus aurantiifolia* (Christm.) Swingle	Cultivated (Not native to Jamaica. Native to Asia)	Common cold and flu (anecdotal)	Leaf (Fresh or dried)	Decoction made 5–6 leaves in 1 cup of water. Steep for ~10 min. Sweeten as desired.	The antiviral mechanism of action is unknown.
21.	Jack-in-the-bush (*Eupatorium odoratum* L.)	Cultivated (Not native to Jamaica.	Common cold (anecdotal)	Leaf (Fresh or dried) and stem	Decoction made of 1–2 tsp of dried leaves brewed in 1 cup of water. Steep for ~10 min. Sweeten as desired.	The antiviral mechanism of action is unknown.

The effective dosage of each medicinal plant preparation is currently unknown. There is not enough rigid scientific research to recommend a standard dosage for any medical condition. Anecdotal evidence recommends one to two cups of oral administration of these decoctions per day, until symptom relief is experienced.

## 11. Toxic Compounds and Adverse Side-Effects

A limitation to homemade decoctions and infusions is due to the fact that, water, regarded as a universal solvent, will, in addition to dissolving plant material, also dissolve unwanted material such as contaminants and toxic compounds. These may have adverse side-effects if ingested. On this tangent, it is of importance that herbs used to make decoctions are grown in a suitable environment. Table 3 below lists some toxic compounds and adverse side-effects associated with frequently used medicinal plants in Jamaica.

## 12. Economic Analysis of Phytomedicines Related to Antiviral Drugs

According to a report titled *“Anti-Viral Drug Therapy Global Market Report 2020–30: COVID-19 Implications and Growth”* by Research and Markets, the global antiviral drug therapy market is expected to grow from USD 52.2 billion in 2019 to about USD 59.9 billion in 2020 [91]. This is primarily due to the increase in demand for antiviral drugs for the treatment of COVID-19. The market is expected to stabilize and reach USD 62.6 billion at a compound annual growth rate (CAGR) of 4.6% through 2023 [91].

According to a report titled *“Global Herbal Medicine Market Research Report—Forecast To 2023”* by Market Research Future, the herbal medicine market is expected to reach USD 111 billion by the end of 2023 [91]. The global herbal medicine market is projected to grow at a compound annual growth rate (CAGR) of ~7.2 % from 2017 to 2023 [92]. 

### Economic Analysis of Phytomedicines in the Caribbean

Jamaica’s medicinal cannabis market value is estimated to be around USD 300–400 million. In consideration of the wide diversity of other Jamaican medicinal plants, this value could increase three-fold. Table 4 directly below lists some valuable patents on Jamaican medicinal plants.

## 13. Conclusions and Future Prospects

As estimated by the World Health Organization (WHO), over 65% of the world population utilize botanical preparations as medicine [1]. Natural alternatives are increasingly gaining attention because of greater accessibility to medicinal plants, and the possibility that they may have fewer and less adverse side-effects than synthetic drugs. This makes natural alternatives more desirable as novel drug therapies. Thousands of plant species still remain to be screened for their bioactivity. Only approximately 15% of some 250,000 species of higher plants have been studied for their pharmaceutical potential [4]. Only 5% of these plants had one or more forms of biological activity [101]. This means that there is an enormous, untapped potential for the development of a myriad of plant-based therapeutics and pharmaceutical-grade proteins likes vaccines, hormones, antibodies and cytokines [102].

Jamaica is home to many established medicinal plants such as ginger, garlic, medina, ganja, cerasee, ball moss and fever grass [103]. This makes the island particularly welcoming for rigorous scientific research on the medicinal value of plants and the development of phytomedicine thereof. This could have great economic and medicinal implications, not only for Jamaica, but for the region. Our traditional use of these medicinal plants in Jamaica can be attributed to the passing of knowledge from traditional Asian and African medicine.

Jamaica’s enormous “ethnopharmacopoeia” in addition to rigorous community-driven scientific data sharing through globalization could potentially expedite natural-product development, lower the cost to purchase natural-products and the cost for commercialization of natural-based drugs, and increase the availability of natural-based pharmaceuticals. Despite the increasing scientific evidence supporting the medicinal value of plants native to and cultivated in Jamaica, commercialization of natural-products derived from these efforts is limited by factors including inadequate testing systems, uncertain regulatory landscapes in natural-product biomedical research, manufacturing practices, commercialization, licensing and intellectual property. As a result, the lack of a rational approach to a creating a sustainable natural-product pharmaceutical industry has been limited.

Modern research around the world should now focus on the pharmacological activities of the phytochemicals and mapping their genomes and transcriptomes to produce target drugs. There is a need for more systematic botanical, physicochemical and chemical analyses of thousands of potential medicinal plants. Further research is required to determine the efficacy, dosage standards, optimum extraction methods/solvents, cytotoxicity/hepatoxicity, pharmacokinetics, molecular mechanisms of action, phytoantiviral screening methods, and drug interactions for many phytoantivirals.

Creating and using universal sets of primers, databases and standards to catalogue species by research groups all around the world, would increase the level of reliability and the number of species available for study (which has reached the greatest level ever achieved by the scientific community) while also making it possible to identify an ever-growing number of species [104]. In addition, a greater awareness of efficacies, toxicities and drug-to-drug interactions of traditional herbal medicines is required. Most importantly, before these plant-based drugs can be incorporated into conventional medicine, more randomized, double-blind, placebo-controlled clinical trials are need on a larger scale. Data generated from such experiments should also be logged in a universal, open-access database. Countries dependent on traditional, synthetic medicine, need to re-shift focus to programs aimed at systematically studying the bioactive compounds from plants, and synthesizing new drugs from compounds.

## Figures and Tables

**Figure 1 molecules-26-00607-f001:**
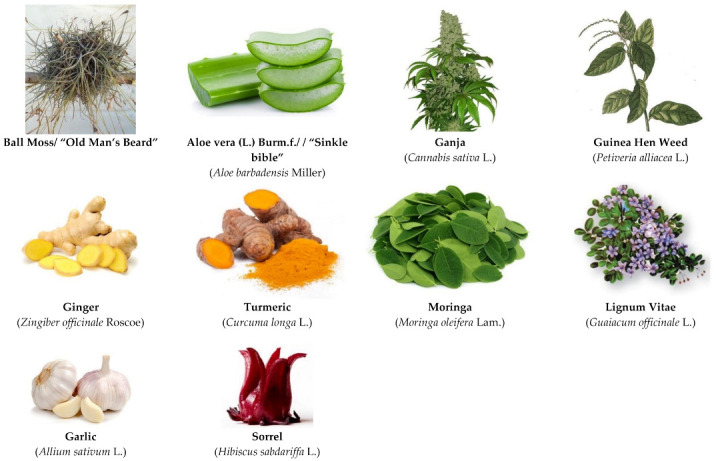
Some Jamaican plants with major antiviral activity.

**Figure 2 molecules-26-00607-f002:**
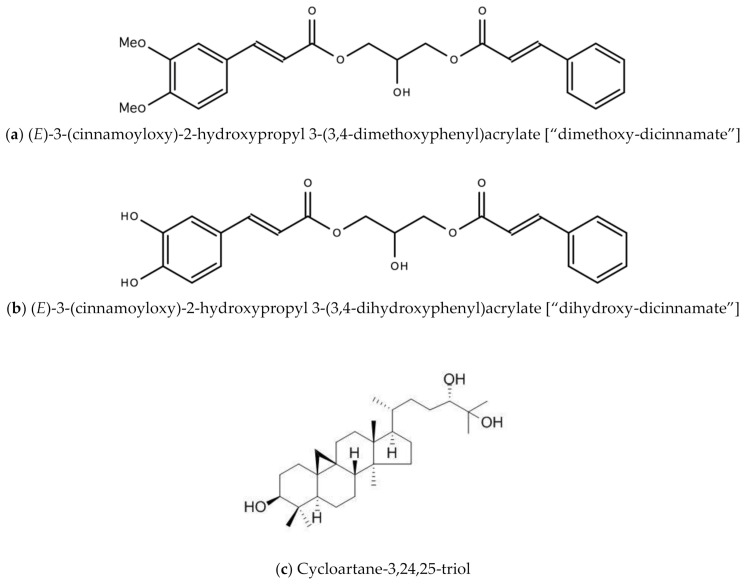
Chemical structures of some bioactive molecules found in *Tillandsia recurvata* L.

**Figure 3 molecules-26-00607-f003:**
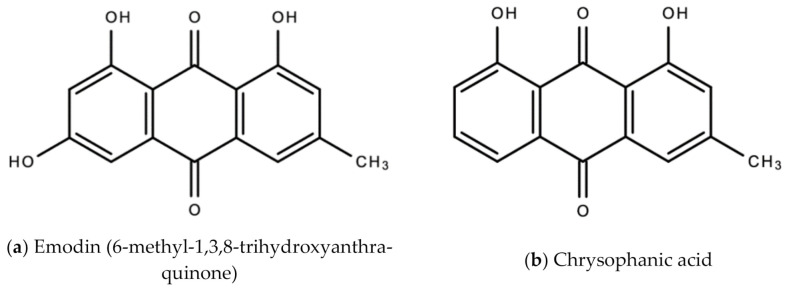
Chemical structures of some bioactive molecules found in *A. vera*.

**Figure 4 molecules-26-00607-f004:**
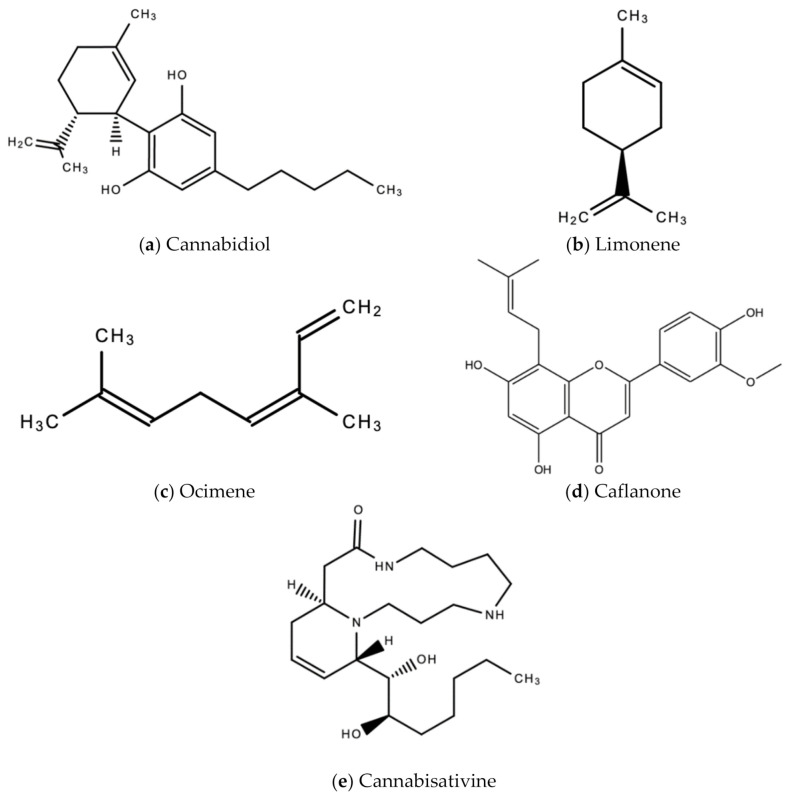
Chemical structures of some bioactive molecules found in *C. sativa* L.

**Figure 5 molecules-26-00607-f005:**
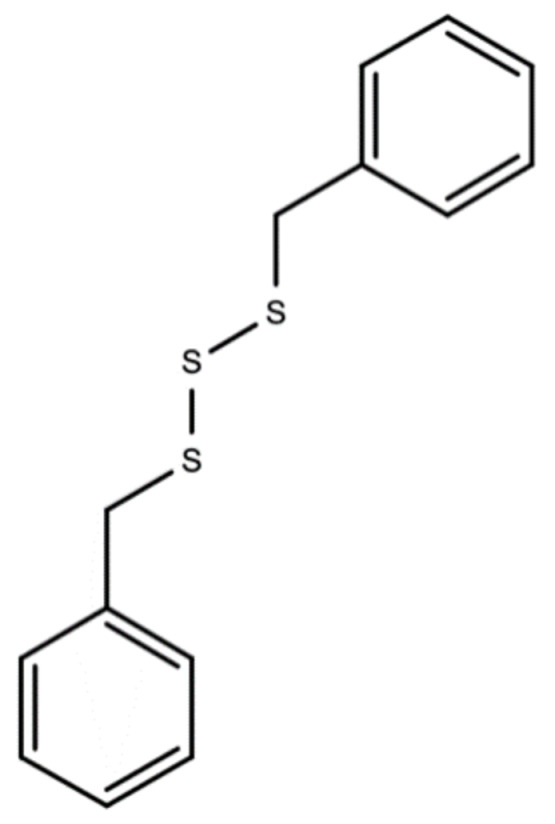
Chemical structures of dibenzyl trisulfide (DTS) found in Guinea Hen Weed.

**Figure 6 molecules-26-00607-f006:**
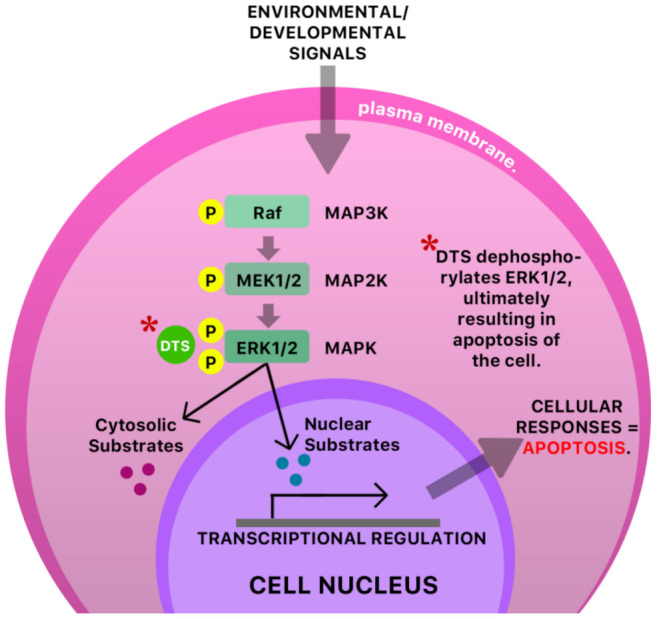
A possible antiviral mechanism of DTS in an infected cell. The mitogen-activated protein kinase/extracellular signal-regulated kinase (MAPK ERK1/ERK2 pathway) is a signaling pathway that is responsible for the transduction of protein components (chemical signals, transcription factors), MAP kinase kinase (MAP2K), and a final kinase, MAP kinase (MAPK), leading to the functioning and regulation of multiple cellular processes like cell proliferation, cell growth, and cell survival. A dysregulation of this pathway or any of its components typically has pathological consequences [43,44]. It has been proposed that some pathological conditions are characterized by increased phosphorylation of kinases [45], possibly resulting in over-proliferation of cells. Dibenzyl trisulfide (DTS) is shown to inhibit the MAPK ERK1/ERK2 pathway in cancer- and (possibly) virally infected cells via dephosphorylation of ERK1/2, ultimately resulting in apoptosis of the cell [45]. MAP—mitogen-activated protein; K—kinase; Raf—rapidly accelerated fibrosarcoma; MEK—mitogen-activated protein kinase.

**Figure 7 molecules-26-00607-f007:**
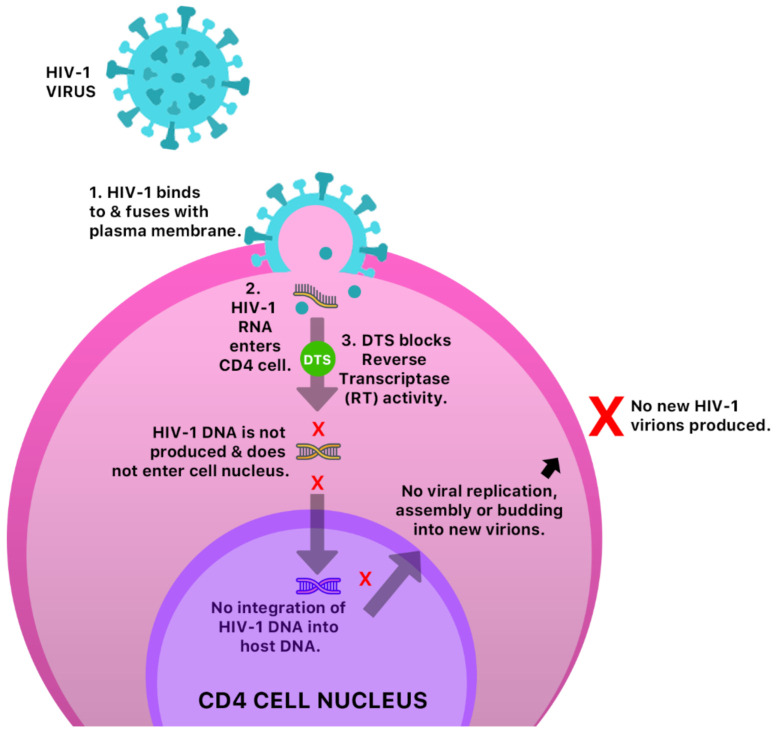
A possible anti-HIV-1 mechanism of DTS in a CD4 cell.

**Figure 8 molecules-26-00607-f008:**
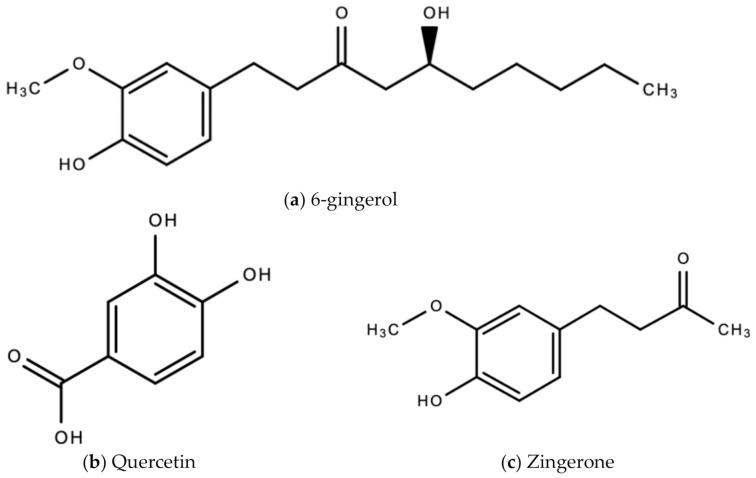
Chemical structures of some bioactive molecules found in *Zingiber officinale* Roscoe.

**Figure 9 molecules-26-00607-f009:**
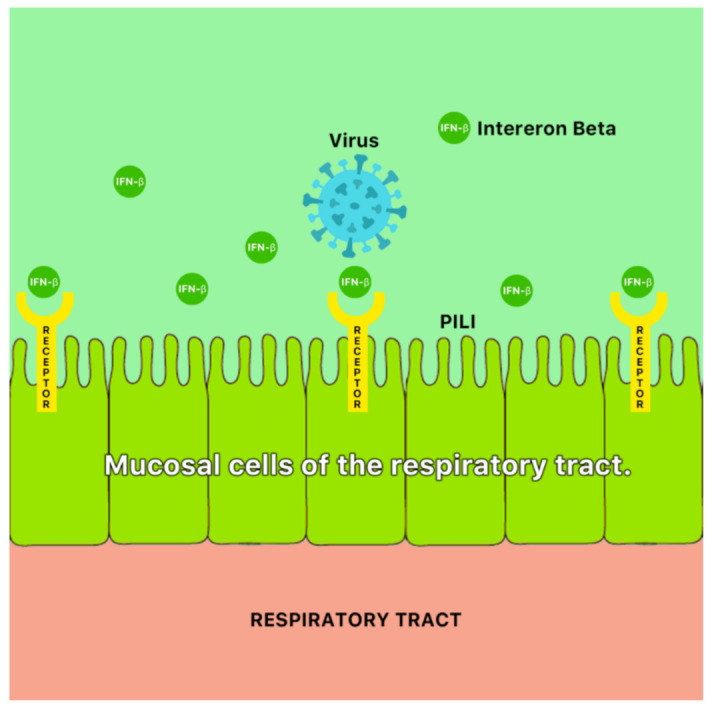
A possible antiviral mechanism of ginger (*Zingiber officinale* Roscoe). It is possible that ginger blocks viral attachment and internalization via stimulation of mucosal cells of the respiratory tract to secrete interferon-β (IFN-β) [46].

**Figure 10 molecules-26-00607-f010:**
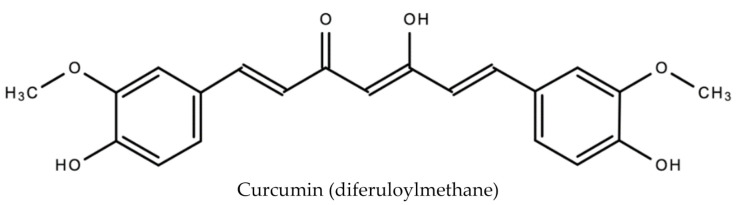
Chemical structure of curcumin (diferuloylmethane) found in *Curcuma Longa L.*

**Figure 11 molecules-26-00607-f011:**
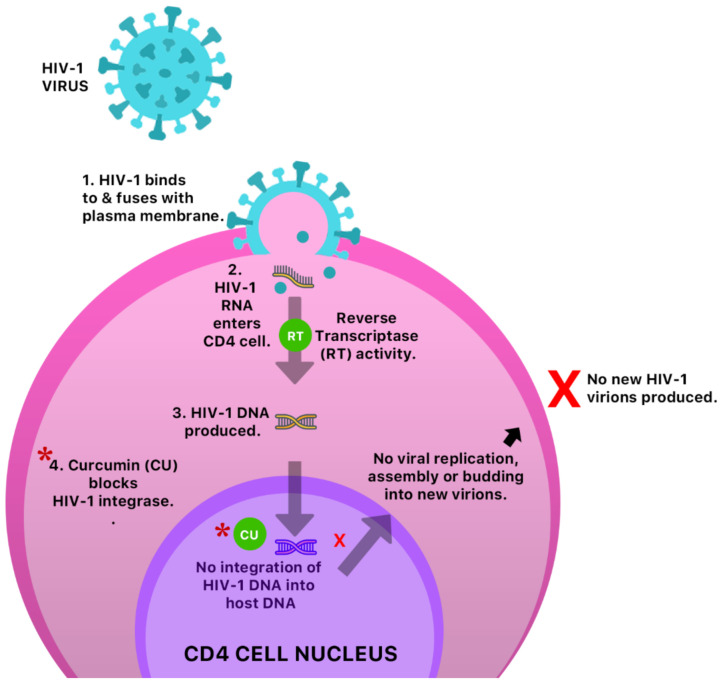
A possible anti-HIV-1 mechanism of curcumin by inhibition of HIV-1 integrase. This prevents integration of HIV-1 DNA into host cell DNA and ultimately inhibition of viral replication, assembly, budding and infection of new cells.

**Figure 12 molecules-26-00607-f012:**
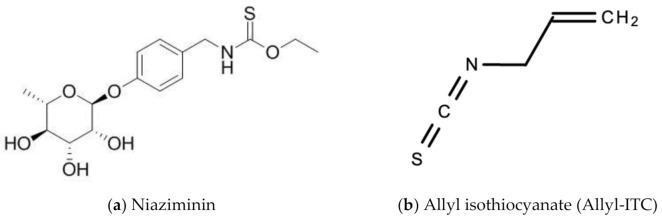
Chemical structures of some bioactive molecules found in *Moringa oleifera* Lam.

**Figure 13 molecules-26-00607-f013:**
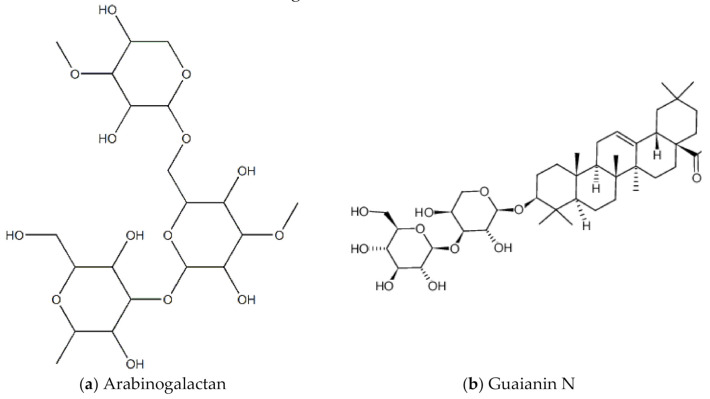
Chemical structures of some bioactive molecules found in Lignum Vitae.

**Figure 14 molecules-26-00607-f014:**
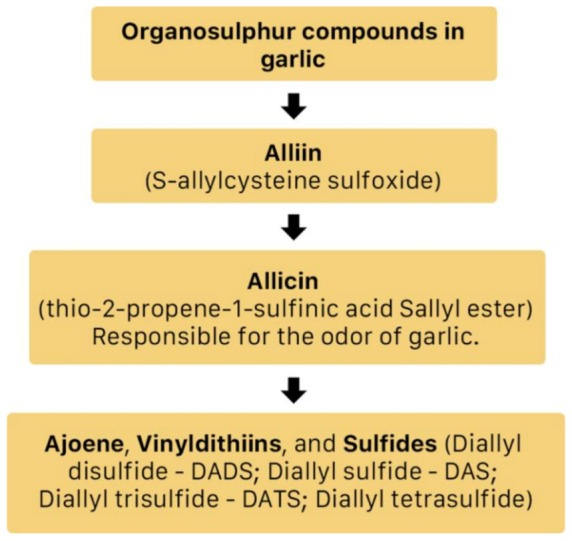
The conversion of organosulfur compounds in garlic [64].

**Figure 15 molecules-26-00607-f015:**
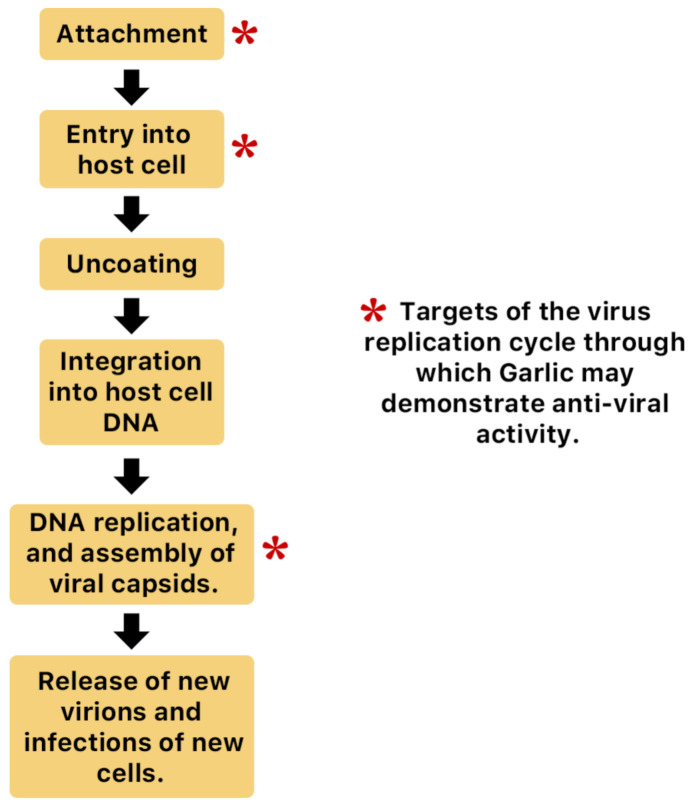
Possible anti-influenza mechanisms of garlic. Organosulfur compounds produced by garlic may inhibit various stages of the general virus life cycle including viral attachment, entry and multiplication [65]. Another possible mechanism of action is via inhibition of components of viral signaling pathways [65].

**Figure 16 molecules-26-00607-f016:**
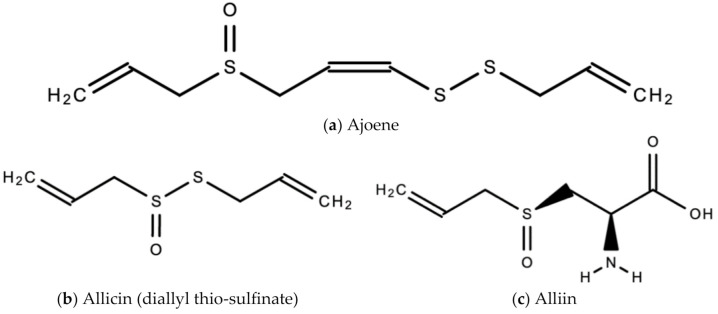
Chemical structures of some bioactive molecules found in garlic.

**Figure 17 molecules-26-00607-f017:**
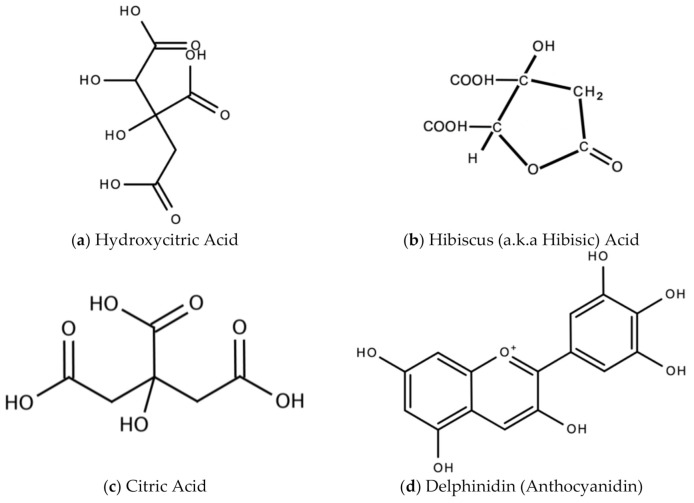
Chemical structures of some bioactive molecules found in sorrel.

**Table 1 molecules-26-00607-t001:** Drugs approved by the FDA between 1 January 1981 and 30 September 2019 that were either derived from botanical drugs, unaltered natural products, or synthetic drugs with natural pharmacophore.

Type of Drug	Number Approved
Natural products	71
Botanical drugs (defined mixture)	14
Natural product mixture	356
Synthetic drugs with natural pharmacophores	207
TOTAL	648

**Table 3 molecules-26-00607-t003:** Some toxic compounds and adverse side-effects associated with frequently used medicinal plants in Jamaica.

	Plant	Possible Contaminants	Type of Medicinal Preparation	Side-Effects
1.	Ginger (*Zingiber officinale* Roscoe)	Mycotoxins, Aflatoxins Ochratoxin A (OTA). Microbes. Chemical residues (pesticides)	Decoction	Generally safe and negligible side effects like bloating, nausea, upset stomach, heartburn, diarrhea, dizziness
2.	Turmeric (*Curcuma longa* L.)	Lead, Chromium, Lead chromate (often used as a yellow coloring). Microbes. Chemical residues (pesticides)	Decoction	Generally safe and negligible side effects like bloating, nausea, upset stomach, heartburn, diarrhea, dizziness
3.	Ball Moss/ “Old Man’s Beard” (*Tillandsia recurvata* L.)	Various contaminants because of the plant’s bioremediatory properties. Dependent on growth environment. Microbes. Chemical residues (pesticides).	Decoction	Generally safe and negligible side effects like bloating, nausea, upset stomach, diarrhea, and dizziness.
4.	*A. vera* (*Aloe barbadensis* Miller)	Various heavy metals like copper, lead, Chromium, mercury, nickel, arsenic, and cadmium. Microbes. Chemical residues (pesticides). *A. vera* is a phytoremediator/ biosorbent [77].	Decoction or Juice	Generally safe and negligible side effects like bloating, nausea, upset stomach, laxative effects, dehydration, lowers sugar and potassium levels, weakness, fatigue, diarrhea, kidney issues, hypersensitivity reactions, and irregular heartbeat [78,79].
5.	*Ganja* (*Cannabis Sativa* L.)	Various contaminants because of the plant’s biosorbent properties. These include heavy metals like cadmium, copper lead and mercury [80,81], residual chemical solvents, pesticides and microbes like Aspergillus that produce mycotoxins. Chemical residues (pesticides).	Decoction	Generally safe and negligible side effects like bloating, nausea, upset stomach, lowers blood pressure, increased heart rate, dizziness, light-headedness, and mild psychoactive effects [82].
6.	Guinea Hen Weed (*Petiveria alliacea* L.)	Chemical residues (pesticides) and microbes. Coumadin—a blood thinner [83].	Decoction	Generally safe and negligible side effects like bloating, nausea, upset stomach, diarrhea, and dizziness. Abortive and hypoglycemic effects [83].
7.	Moringa (*Moringa oleifera* Lam.)	Various contaminants because of the plant’s biosorbent properties. These include heavy metals like cadmium, copper lead and mercury and microbes [84]. Chemical residues (pesticides)	Decoction	Generally safe and negligible side effects like bloating, nausea, upset stomach, lowers blood pressure and heart rate, fertility interference, uterine contractions [85].
8.	*Lignum Vitae (Guaiacum officinale* L.)	Chemical residues (pesticides) and microbes.	Decoction	Generally safe and negligible side effects like bloating, nausea, upset stomach.
9.	Garlic (*Allium sativum* L.)	Chemical residues (pesticides). Microbial contaminants [86] and heavy metals like lead and sulfites [87].	Decoction	Generally safe and negligible side effects like bloating, nausea, gas, upset stomach, bad breath and body odor, vomiting, diarrhea, heartburn, mouth/throat burn [88].
10.	Sorrell (*Hibiscus sabdariffa* L.)	Chemical residues (pesticides) and microbes.	Decoction	Generally safe and negligible side effects like bloating, nausea, upset stomach, diarrhea, and dizziness.
11.	“Search-mi-heart” (*Rhytidophyllum tomentosum* (L.) Mart.	Chemical residues (pesticides) and microbes.	Decoction	Generally safe and negligible side effects like bloating, nausea, upset stomach, diarrhea, and dizziness.
12.	Pepper elder (*Piper amalago* L.)	Chemical residues (pesticides) and microbes.	Decoction	Generally safe and negligible side effects like bloating, nausea, upset stomach, diarrhea, and dizziness.
13.	McKatty Weed/Marigold (*Bidens reptans* (L.) G.Don	Chemical residues (pesticides) and microbes.	Decoction	Generally safe and negligible side effects like bloating, nausea, upset stomach, diarrhea, and dizziness.
14.	Chany Root (*Smilax balbisiana* Kunth)	Chemical residues (pesticides) and microbes.	Decoction	Generally safe and negligible side effects like bloating, nausea, upset stomach, diarrhea, and dizziness.
15.	Lemongrass/Fevergrass (*Cymbopogen citratus* L.)	Chemical residues (pesticides) and microbes.	Decoction	Generally safe and negligible side effects like bloating, nausea, upset stomach, diarrhea, and dizziness.
16.	Cerassee (*Momordica charantia* L.)	Chemical residues (pesticides) and microbes.	Decoction	Generally safe and negligible side effects like bloating, nausea, upset stomach, diarrhea, and dizziness.
17.	Soursop (*Annona muricata* L.)	Chemical residues (pesticides) and microbes. Annonacin, an acetogenin that is toxic to the nervous system [89].	Decoction	Generally safe and negligible side effects like bloating, nausea, upset stomach, movement disorders, sensation issues [90]. Consumption = Increased risk of atypical parkinsonism development [89].
18.	Leaf of life (*Bryophyllum pinnatum* (Lam.) Oken	Chemical residues (pesticides) and microbes.	Decoction	Generally safe and negligible side effects like bloating, nausea, upset stomach, diarrhea, and dizziness.
19.	Dogblood (*Rivina humilis* L.)	Chemical residues (pesticides) and microbes.	Decoction	Generally safe and negligible side effects like bloating, nausea, upset stomach, diarrhea, and dizziness.
20.	Lime Leaf Tea (*Citrus aurantiifolia* (Christm.) Swingle	Chemical residues (pesticides) and microbes.	Decoction	Generally safe and negligible side effects like bloating, nausea, upset stomach, diarrhea, and dizziness.
21.	Jack-in-the-bush (*Eupatorium odoratum* L.)	Chemical residues (pesticides) and microbes.	Decoction	Generally safe and negligible side effects like bloating, nausea, upset stomach, diarrhea, and dizziness.

**Table 4 molecules-26-00607-t004:** Examples of patents on Jamaican Medicinal Plants.

	Therapeutic Window/Benefits of Flavonoids	Plant	Patent Number	Reference
1.	Therapeutic agents containing cannabis flavonoid derivatives targeting kinases, sirtuins and oncogenic agents for the treatment of cancers.	*Cannabis sativa* L.	US20180098961A1	[93]
2.	Agent containing flavonoid derivatives for treating cancer and inflammation.	*Cannabis sativa* L.	US20170360744A1	[94]
3.	Therapeutic agents containing cannabis flavonoid derivative for ocular disorders.	*Cannabis sativa* L.	US10278950B2	[95]
4.	Therapeutic agents containing cannabis flavonoid derivatives for the prevention and treatment of neurodegenerative disorders.	*Cannabis sativa* L.	US10751320B2	[96]
5.	Pi 4-kinase inhibitor as a therapeutic for viral hepatitis, cancer, malaria. autoimmune disorders and inflammation, and a radiosensitizer and immunosuppressant.	*Vernonia acuminata* DC.	WO2018022868A1	[97]
6.	Therapeutic antiviral agents containing cannabis cannabinoid derivatives.	*Cannabis sativa* L.	US20180214389A1	[98]
7.	Methods for inhibiting HIV-1 activity by inhibitory mechanisms of extracts of *Guaiacum officinale* L. (*Zygophyllaceae*)	*Guaiacum officinale* L.	US9814747B2	[99]
8.	Therapeutic potential of dibenzyl trisulfide Isolated from *Petiveria alliacea* L. (Guinea Hen Weed, Anamu)	*Petiveria alliacea* L.	Patent pending	[45,100]

## Data Availability

Data sharing is not applicable to this article. No new data were created or analyzed in this study.

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
