# Peer review of "Antiviral Activity of Jamaican Medicinal Plants and Isolated Bioactive Compounds"

_molecules, 2021, doi:10.3390/molecules26030607_

Round 1
Reviewer 1 Report
This is a review on the Jamaican Medicinal Plants with Special Reference to Their Antiviral Activity which lacks a clear focus and fails to offer new information to the reader interested in the wealth of medicinal plants in Jamaica that are potentially antiviral. Although the authors report ethnobotanical/ethnopharmacological studies in the Caribbean (ref. 13 and 15 -probably and 14 but it is not referred in the text), they do not actually use them. I would expect that they would select the Jamaican plants that are used against viral infections and and present more info on them, i.e. whether they are native or cultivated, the plant parts used and the way of preparation, if we know the composition, regardless of whether these are tested in lab or not. Instead, the authors provide info on plants which are either cultivated worldwide or are characteristic of other places in the world like cannabis, garlic, ginger, turmeric, moringa and Hibiscus. For instance, turmeric and Moringa are characteristic of Southeast Asia. Thus, my main feeling is that they do not present novel information about Jamaica and present common knowledge. I would be convinced if they sufficiently justified why they present information on plants they select, e.g. on ethnopharmacological reasons and what is their connection with Jamaica (they are native, cultivated or imported). Moreover, the way the information is presented is not complete and appropriate fo a journal entitled Molecules, since one would expect more information on the bioactives (there is no chemical structure) and on the plant parts (roots, leaves, the whole plant?) from which each bioactive phytochemical are extracted.
Other major points:
- All the plant names are not written in a correct acceptable way according to international standards. The authors should check all plant names in an acceptable database like The Plants' list (http://www.theplantlist.org) and select the accepted name, present the synonyms and write them correctly in the first occurrence, e.g. Aloe vera (L.) Burm.f. (synonym: Aloe barbadensis Mill.). Later on, they can use the abbreviation, e.g. A. vera.
- A major argument of the authors is that the herbal drugs "have fewer and less adverse side-effects, are safer, and have potentially greater therapeutic efficacy" (lines 41-42, 436-437). I strongly object and I would urge the authors not to be so absolute on this point (except they provide arguments), since many poisons have been derived plants (e.g. aristolochic acid which they mention), drugs that are toxic in high doses (e.g. cardiac glycosides) and many herbal drugs have important side effects and drug interactions (e.g. St. John's wort).
- Lines 218-232. Appropriate references are necessary for the potential therapeutical usefulness of antioxidant molecules in antiviral therapy. As it is, it fails to be convincing.
- Minor points
- lines 35-40. Please look again the syntax of the sentence. Infections are caused by viruses and are not viruses.
- lines 57-58: on which year?
- line 65: estimated
- line 70: disrupt
- lines 89 and 93. Please provide links for the databases.
- line 98: delete in
- line 105: what do you mean? Which strains?
- line 107. comma after pain.
- line 187: is hypericin present in Aloe? Please double check.
- Lines 202 and 203. "around 545" what? Please rewrite.
- line 208. Have alkaloids been described in cannabis? Reference?
- line 283: I guess "are" should be deleted.
- line 370-371. Double check and rewrite providing more information. Arabinogalactans are usually not disaccharides and not phenolic. Which saponins?
- line 464: delete said.
Author Response
Reviewer 1
This is a review on the Jamaican Medicinal Plants with Special Reference to Their Antiviral Activity which lacks a clear focus and fails to offer new information to the reader interested in the wealth of medicinal plants in Jamaica that are potentially antiviral. Although the authors report ethnobotanical/ethnopharmacological studies in the Caribbean (ref. 13 and 15 -probably and 14 but it is not referred in the text), they do not actually use them. I would expect that they would select the Jamaican plants that are used against viral infections and present more info on them, i.e. whether they are native or cultivated, the plant parts used and the way of preparation, if we know the composition, regardless of whether these are tested in lab or not. Instead, the authors provide info on plants which are either cultivated worldwide or are characteristic of other places in the world like cannabis, garlic, ginger, turmeric, moringa and Hibiscus. For instance, turmeric and Moringa are characteristic of Southeast Asia. Thus, my main feeling is that they do not present novel information about Jamaica and present common knowledge. I would be convinced if they sufficiently justified why they present information on plants they select, e.g. on ethnopharmacological reasons and what is their connection with Jamaica (they are native, cultivated or imported). Moreover, the way the information is presented is not complete and appropriate for a journal entitled Molecules, since one would expect more information on the bioactives (there is no chemical structure) and on the plant parts (roots, leaves, the whole plant?) from which each bioactive phytochemical are extracted.
Response:
- Table 2 clarifies the nativity of plants. Table 2 also lists:
- Methodology of medicinal preparation.
- Part of the plant used to make medicinal preparation/the plant parts from which the bioactives are extracted.
- Bioactive constituents.
- Possible antiviral mechanisms of action.
- Figures of the chemical structures of examples of biomolecules from each medicinal plant have been included.
- May you please clarify the issues with references 13, 14 and 15?
- Yes, some of these medicinal plants are not unique to Jamaica, nor are the methodologies of medicinal preparations. They are made mention of in this article, only because they are frequently used in Jamaica for their antiviral activity.
Other major points:
- All the plant names are not written in a correct acceptable way according to international standards. The authors should check all plant names in an acceptable database like The Plants' list (http://www.theplantlist.org) and select the accepted name, present the synonyms and write them correctly in the first occurrence, e.g. Aloe vera (L.) Burm.f. (synonym: Aloe barbadensis Mill.). Later on, they can use the abbreviation, e.g. A. vera.
Response:
Correction of scientific names of plants and abbreviations.
- A major argument of the authors is that the herbal drugs "have fewer and less adverse side-effects, are safer, and have potentially greater therapeutic efficacy" (lines 48-49, 600-602). I strongly object and I would urge the authors not to be so absolute on this point (except they provide arguments), since many poisons have been derived plants (e.g. aristolochic acid which they mention), drugs that are toxic in high doses (e.g. cardiac glycosides) and many herbal drugs have important side effects and drug interactions (e.g. St. John's wort).
Response:
I am not so absolute on this point. I could not find the reference to this claim. As a result, this sentence has been rephrased. Table 2 shows:
- Methodology of medicinal preparation.
- Part of the plant used to make medicinal preparation.
- Bioactive constituents.
- Possible antiviral mechanisms of action.
Table 3 shows possible contaminants found in plants and possible side-effects from consumption of decoctions/juices made from these medicinal plants.
- Lines 218-232. Appropriate references are necessary for the potential therapeutical usefulness of antioxidant molecules in antiviral therapy. As it is, it fails to be convincing.
Response:
Minor points
- Lines 35-40. Please look again the syntax of the sentence. Infections are caused by viruses and are not viruses.
Response:
Syntax of line changed. “Viral infections caused by the…”(Line 32-34)
- lines 57-58: on which year?
Response:
This comment is unclear. May you please clarify.
- line 65: estimated
Response:
“Estimate” à “estimated” (Line 91)
- line 70: disrupt
Response:
“Disrupts” changed to “disrupt” (Line 97)
- Line 117. Please provide links for the databases.
Response:
Sentence restructured. Reference provided (Line 117)
- line 98: delete in
Response:
The word “in” removed (Line 127)
- line 105: what do you mean? Which strains?
Response:
The word “strains” is changed to “medicinal plants”, and “biodiversity” à “diversity” (Line 127)
- line 107. comma after pain.
Response:
Comma after “pain” (line 129).
- line 187: is hypericin present in Aloe? Please double check.
Response:
I was able to confirm that Hypiricin is not present in A. vera, but instead in Hypericum perforatum L., known as St. John's Wort of the Hypericaceae species. This has been corrected.
- Lines 202 and 203. "around 545" what? Please rewrite.
Response:
Line changed to “It is estimated that C. sativa produces an estimated 545 chemical compounds…”. Full-stop removed after “545” (Line 270).
- line 208. Have alkaloids been described in cannabis? Reference?
Response:
Alkaloids have been described in cannabis. Please see reference #28.
Mechoulam, R. (1989). Chapter 2 Alkaloids in Cannabis Sativa L. The Alkaloids: Chemistry and Pharmacology, 77-93. doi:10.1016/s0099-9598(08)60227-8
- line 283: I guess "are" should be deleted.
The word “are” is changed to “is”. Line now reads “In moderation, ROS is involved in homeostasis and signaling and is associated…” (Line 296)
- line 370-371. Double check and rewrite providing more information. Arabinogalactans are usually not disaccharides and not phenolic. Which saponins?
Response:
The reference about arabinogalactans could not be found, so the sentence was removed. Types of saponins in Guaiacum officinale L have been listed (Line 490).
- line 464: delete said.
Responses:
The word “said” was removed (Line 642).
Reviewer 2 Report
The manuscript “Important Jamaican Medicinal Plants with Special Reference to Their Antiviral Activity”, by Henry Lowe et al. is of interest. However, some part of the review article needs, in my opinion, of further work and changes in organization of the chapters included in the paper.
Major points:
- The Title does not reflect the content of the article. The article is also focused on the compounds derive from the considered medicinal plant. I suggest to include the term “compounds” (or molecules, or agents or similar) in the Title. Also, the term “important” might be eliminated (a possible Title might be: “Antiviral Activity of Jamaican Medicinal Plants and isolated bioactive compounds”).
- A chapter is missing about the part of the plant that are employed to generate “medicinal preparations” (Leaves? Roots? Other part). I would suggest to include a short chapter on this issue, just to clarify to the reader that the authors are not talking about “total plant extracts”.
- Can the authors comment about the sustainability of the industrial production of Jamaican Medicinal Plants that exhibit antiviral applications in the case of wide-spread pathologies? There are, in the authors opinion, problems related to the sustainability on one hand and to the possible alteration of the biodiversity on the other?
- Some of the ingredients of medicinal plants are clearly toxic (for instance DNA intercalating compounds). I suggest to include a short chapter describing reported side effects of the medicinal plants considered in the review.
- A Table including representative example of patents on Jamaican Medicinal plants might be useful to verify the industrial interest.
- Throughout the manuscript (I am referring to chapters 3.1-3.10) the information about un-fractionated plant extracts and isolated ingredients is mixed. In my opinion it would be interesting to separate the effects of “extracts” from the effects of “isolated compounds” in two separate sub-chapters.
Minor points
- Line 38. COVID-19 is coronavirus disease: please implement, if you agree.
- Please add the references to Figure 1 relative to the cited applications.
- The reference list might be implemented
Author Response
The manuscript “Important Jamaican Medicinal Plants with Special Reference to Their Antiviral Activity”, by Henry Lowe et al. is of interest. However, some part of the review article needs, in my opinion, of further work and changes in organization of the chapters included in the paper.
Major points:
- The Title does not reflect the content of the article. The article is also focused on the compounds derive from the considered medicinal plant. I suggest to include the term “compounds” (or molecules, or agents or similar) in the Title. Also, the term “important” might be eliminated (a possible Title might be: “Antiviral Activity of Jamaican Medicinal Plants and isolated bioactive compounds”).
Response:
Title changed to “Antiviral Activity of Jamaican Medicinal Plants and isolated bioactive compounds”).
- A chapter is missing about the part of the plant that are employed to generate “medicinal preparations” (Leaves? Roots? Other part). I would suggest to include a short chapter on this issue, just to clarify to the reader that the authors are not talking about “total plant extracts”.
Response:
A table listing how medicinal preparations has been included (table 2).
- Can the authors comment about the sustainability of the industrial production of Jamaican Medicinal Plants that exhibit antiviral applications in the case of wide-spread pathologies? There are, in the authors opinion, problems related to the sustainability on one hand and to the possible alteration of the biodiversity on the other?
Response:
Commentary has been provided on the sustainability of the industrial production of Jamaican medicinal plants that exhibit antiviral applications in the case of widespread pathologies. Commentary has also been provided on the ethnopharmacological connection of these plants to Jamaica. (Paragraph starting at line # 615).
- Some of the ingredients of medicinal plants are clearly toxic (for instance DNA intercalating compounds). I suggest to include a short chapter describing reported side effects of the medicinal plants considered in the review.
Response:
A table (table 3) listing examples of the following have been included:
- Toxic compounds found in medicinal plants
- Reported side-effects of medicinal plants
- A Table including representative example of patents on Jamaican Medicinal plants might be useful to verify the industrial interest.
Response:
A table (table 4) listing examples of patents on Jamaican medicinal plants has been included.
- Throughout the manuscript (I am referring to chapters 3.1-3.10) the information about un-fractionated plant extracts and isolated ingredients is mixed. In my opinion it would be interesting to separate the effects of “extracts” from the effects of “isolated compounds” in two separate sub-chapters.
Minor points
- Lines 281 and 354. COVID-19 is coronavirus disease: please implement, if you agree.
Responses:
Yes, COVID-19 is human coronavirus hCov-OC43 disease. This has been clarified throughout the text.
- Please add the references to Figure 1 relative to the cited applications.
Response:
References have been included in Figure 1.
- The reference list might be implemented
Response:
References have been included in Figure 1.
Reviewer 3 Report
In the present work are reviewed Jamaican medicinal plants that have bioactive potential as the focus is on plants with anti-viral activities including anti-Covid-19. The review gives information about plant metabolites and their activities and points out strategies for utilization of medicinal plants as drug alternatives. This is a current topic and the information about Jamaican medicinal plants is very interesting. The text is clearly written and structured. The figures are well visualized and comprehensive.
Author Response
No revisions suggested.
Round 2
Reviewer 1 Report
The revised article addressed all comments raised in the first review. However, the Tables now provided are difficult to read so improvements in the layout are suggested. Moreover, the column "Some Bioactive Constituents" in Table 2 could be omitted since this information is later on presented in detail and with the appropriate references (some of which are missing in the Table). The authors did check the correct scientific names of the plants but in many cases it is written wrong (I mean which part is written in italic and which is not), e.g. Allium sativum L. instead of Allium sativum L. Section 4.1 (Terpenoids...) seems out of place since the general category in 4 is Plants. The authors could delete it. Moreover, look carefully Figure 15 because within the figure it says ginger althought the figure refers to garlic. I believe that the article still has some generalizations and repetitions which could be ommitted so as to make the article shorter and to the points.
line 57: Penicillium chrysogenum is not a plant
line 72: replace "rosmarinic acid" with "phenolic acid derivatives"
line 73: delete alkaloids
line 79: delete "extracts of"
line 230: I believe "amelioration" is better than "induction" (which is misleading)
line 463: Provide justification for the statement "Some 52%... in Jamaica".
Author Response
Moreover, the column "Some Bioactive Constituents" in Table 2 could be omitted since this information is later on presented in detail and with the appropriate references (some of which are missing in the Table).
Response = The column titled “Some bioactive constituents” in table 2 has been omitted. Improvements to the layout have been made to make reading easier.
The authors did check the correct scientific names of the plants but in many cases it is written wrong (I mean which part is written in italic and which is not), e.g. Allium sativum L. instead of Allium sativum L.
Response = Plant names have been correct (un-italicized).
Section 4.1 (Terpenoids...) seems out of place since the general category in 4 is Plants. The authors could delete it.
Response = The section titled “Other Jamaican medicinal plants” has been removed as this was a repetition of information. Now the section on terpenoids now titled “Terpenoids of Cannabis Sativa L. as antiviral agents” falls under the “Ganja (Cannabis sativa L.) category.
Moreover, look carefully Figure 15 because within the figure it says ginger although the figure refers to garlic.
Response = That was a typographical error. That is meant to be “Garlic” instead of “Ginger”. This has been corrected.
I believe that the article still has some generalizations and repetitions which could be omitted so as to make the article shorter and to the points.
Response = Redundant section titled “Other Jamaican medicinal plants” has been removed.
line 57: Penicillium chrysogenum is not a plant
Response = Penicillium chrysogenum has been replaced with “Cannabis sativa L.”
line 72: replace "rosmarinic acid" with "phenolic acid derivatives"
Response = “Rosmarinic acid” has been removed. The sentence now reads “Some active compounds in these plants responsible for the therapeutic effects of phytoantivirals include alkaloids, anthraquinones, coumarins, polyphenols (e.g., flavonoids), phenolic acids and their derivatives, lignans, naphthoquinones, peptides, alkaloids, nitrogenated compounds, polysaccharides and terpenes.”
line 73: delete alkaloids
Response = The word “alkaloids” has been removed.
line 79: delete "extracts of"
Response = The words “extracts of” have been removed.
line 230: I believe "amelioration" is better than "induction" (which is misleading)
Response = The word “induction” has been replaced with “amelioration”.
line 463: Provide justification for the statement "Some 52%... in Jamaica".
Response = Reference not found. The line that reads “Some 52% of the established medicinal plants that exist on earth grow in Jamaica” now reads “Whether endemic or cultivated, Jamaica is home to many established medicinal plants like ginger, garlic, ganja, guinea hen weed, and ball moss”
Reviewer 2 Report
I thank the authors for reviewing the paper introducing changes in agreement with the reviewer's suggestions.
Author Response
Thanks for reviewing the article!